# Problems and Needs Persist for Oklahoma City Bombing Survivors Many Years Later

**DOI:** 10.3390/bs11020019

**Published:** 2021-01-29

**Authors:** Phebe Tucker, Betty Pfefferbaum, Kevin Watson, Landon Hester, Christopher Czapla

**Affiliations:** Department of Psychiatry, University of Oklahoma Health Sciences Center, Oklahoma City, OK 73104, USA; Betty-Pfefferbaum@ouhsc.edu (B.P.); Kevin-Watson@ouhsc.edu (K.W.); Landon-Hester@ouhsc.edu (L.H.); Christopher-Czapla@ouhsc.edu (C.C.)

**Keywords:** posttraumatic growth, posttraumatic stress, resilience, terrorism, mental health, disaster mental health, long term, trauma

## Abstract

Background: This study assesses long-term physical and emotional symptoms and unmet needs in direct survivors of the 1995 Oklahoma City terrorist bombing 18 ½ years after the event. Methods: A telephone questionnaire assessed psychiatric symptoms, health problems and coping strategies in 138 terrorism survivors (of whom 80% were physically injured) from a state registry of directly exposed persons, and 171 non-exposed community controls. Structured survey questions measured psychiatric symptoms, posttraumatic growth, general health problems and health care utilization. Open-ended questions explored survivors’ most important terrorism-related problems and needs. Quantitative and qualitative data analysis methods were undertaken. Results: Survivors reported similar rates of major health problems and general health care utilization, more anxiety and depression symptoms, and more ancillary health care use than controls on structured assessments. Survivors also reported posttraumatic growth, using several positive coping skills. Open-ended questions identified survivors’ specific continuing bombing-related problems, and needs which were not disclosed on the questionnaire; these included many lasting physical injuries, health problems (especially hearing difficulties), specific posttraumatic stress disorder (PTSD) symptoms, other emotional symptoms, work and financial problems, interpersonal issues, and desires to help others. Conclusions: Results suggest that extended recovery services are needed long after terrorism exposure, and that open-ended assessment is useful to identify those requiring services.

## 1. Introduction

The impact of terrorism on survivors has been the focus of numerous international and domestic studies, with more studies assessing short-term effects and fewer assessing long-term emotional and medical sequelae and survivors’ needs related to events [1,2,3].

International acts of terrorism have spanned many continents, affecting numerous directly exposed individuals’ physical and mental health over time. Among Kenyan civilians directly exposed to the 1998 US Embassy bombing in Nairobi, 28% had current posttraumatic stress disorder (PTSD) when assessed three years later, a rate similar to Oklahoma City (OKC) bombing survivors [4]. Survivors of two terrorist attacks in Norway in 2011 who had been hospitalized for physical injuries were assessed three to four years later. A majority had long-term somatic and psychological problems and decreased functioning, with two thirds reporting unsatisfactory physical health. Continuing healthcare and unmet needs were noted by almost all [5]. An annual survey of over 700 survivors of the Tokyo subway sarin attack initiated five years after the event found that the prevalence of somatic symptoms was 60–80% and did not decrease over time. The prevalence of posttraumatic stress was approximately 35% and also did not change over time, suggesting that some survivors needed help for long-term somatic and psychological effects [6].

Devastating acts of domestic terrorism, notably the 1995 Oklahoma City bombing and the terrorist events of 9/11, have had impact on many and have inspired investigations of short- and long-term sequelae in those affected. A large study of family members 14 years after bereavement from the 9/11 attacks found that those with comorbid mental health conditions with no PTSD and those with comorbid mental health problems with PTSD and grief-related impairment were more likely to meet thresholds for depression, grief and anxiety conditions than healthy bereaved participants. Results suggested that clinical programs should focus on high risk individuals to identify those in need of long-term services [7]. A 16-year follow-up study of Ground Zero recovery workers responding to 9/11 identified lasting musculoskeletal injuries and respiratory disorders as well as PTSD, anxiety, depression, insomnia, addictions and risk-taking behaviors [8].

The 1995 OKC bombing killed 168 and injured or bereaved hundreds of persons. It offered a rare opportunity to study survivors’ outcomes over time, as many directly exposed survivors from an Oklahoma State Health Department (OSDH) bombing registry remained within the community and agreed to contribute to future research. Drawing from this registry, studies found a slight decrease in rates of PTSD diagnoses over time. North et al. found one third of direct survivors to have PTSD and nearly one half of survivors to have an active psychiatric disorder after six months, leading to occupational impairment [9]. At 17 months, 31% had current- bombing-related PTSD [10]; by seven years 26% had current PTSD [11]. A telephone survey 18 and a half years post-bombing drawing from the same OSDH registry found that 23.2% of survivors had probable PTSD. Additionally, survivors’ anxiety and depression symptoms (but not PTSD symptoms) were associated with heavy drinking. Survivors had worse mental health and social functioning than demographically matched, non-exposed community controls [12]. Considering long-term mental health service use among highly exposed survivors of the bombing, 6% reported mental health service use over seven years post disaster, with most initiated in the first six months. While 33% received various types of mental health treatments for more than a year, only 7% were receiving services at seven years [13]. Considering medical issues, direct OKC bombing survivors assessed up to three years post disaster had continued hearing problems, worsening of existing medical conditions such as asthma, bronchitis and depression, and increased use of health and mental health services [14]. However, by 18 and a half years post-bombing, direct OKC survivors did not differ from community controls in major medical problems [12].

The current study aims to advance our understanding of long-term problems and needs resulting from direct exposure to terrorism. Such issues may not be captured with questionnaire items that do not assess the cause or severity of medical problems. Open-ended interviews were designed on the original telephone survey to allow survivors to spontaneously report problems and needs that might not fit neatly into categories on questionnaires, or which individuals may recall when allowed to activate memories and respond freely without constraining their responses. Accordingly, our telephone survey 18 and a half years after the OKC bombing was designed to uncover these issues, assessing problems through both questionnaires and open-ended interviews. In this study, we re-examine survivors’ responses to open-ended questions elicited at the same time that structured questions had been asked to compare qualitative with quantitative data. Thus, this study expands our understanding of long-term problems beyond our existing report of enduring problems noted above that used traditional rating scales 18 and a half years post-disaster [12].

## 2. Methods

Invitations to participate in a telephone survey were mailed to 407 survivors directly exposed to the OKC bombing from the OSDH bombing registry who had originally agreed in 1996 to 1998 to participate in future research. A convenience sample of 171 community members not directly exposed to the bombing and without close family or friends killed or injured by the event were recruited through random digit dialing methods to demographically match survivors (frequency matching). Methods for recruiting survivors and community members are further described in a previous publication [12]. Ethical approval for the study was obtained by the institutional review boards of the University of Oklahoma Health Sciences Center (OUHSC) and the OSDH.

### 2.1. Survey Instruments

A 30-min, 100-item telephone survey approved by the above OUHSC and OSDH institutional review boards assessed demographics, general medical status and medical treatments, mental health symptoms and coping mechanisms employed. Hopkins Symptom Checklist (HSCL-25) queried 25 symptoms of psychological problems in general areas of depression and anxiety [15], with reliabilities of 0.88 and 0.93 for anxiety and depression subscales in this study. Breslau’s PTSD screen assessed seven PTSD symptoms, with two from the avoidance and numbing cluster and two from the hyper-arousal cluster; scores of ≥ four symptoms are consistent with PTSD diagnosis by DSM-IV criteria [16]. Reliability statistics (Cronbach α) of Breslau’s PTSD screen was 0.82 in this study. The short form of the Posttraumatic Growth Inventory (PTGI-S) consisted of 10 questions to measure survivors’ use of positive coping skills, such as social support, spiritual beliefs, inner strengths and others [17,18]. Survivors noted whether they “experienced this change as a result of my crisis,” rating from 0 (“did not experience this change as a result of my crisis”) to 5 (experiencing change “to a very great degree”). Reliability of the PTGI-S scale (Cronbach α) was 0.90 in this study.

Questions from the Medical Status Questions (MSQ) used in a prior OSDH survey [11] explored bombing exposure, smoking and alcohol consumption, contacts with health care providers, medical status and the presence of general medical conditions such as stroke, heart disease, hypertension, chronic obstructive pulmonary disease (COPD), diabetes mellitus, cancer and musculoskeletal or inflammatory conditions. Health care providers in areas of physical, speech, respiratory or occupational therapy were considered as ancillary health care workers in this study. At the end of the survey, open-ended questions invited survivors to name the three most important problems they experienced as a result of the bombing and the three most important needs they had at the time of the survey.

### 2.2. Statistical Analyses

Using quantitative methods for structured survey items, independent-samples *t*-test compared means of continuous variables for survivors and controls, with nonparametric Wilcoxon test used when *t*-test was not applicable. Pearson correlation coefficient estimated correlations between continuous variables. Pearson χ^2^ or Fisher exact test compared proportions between the two groups. All analyses were performed with SAS 9.2 (SAS Institute Inc, Cary, NC, USA). The level of statistical significance was set at 0.05.

Using qualitative methods, open-ended questions on unstructured interviews were tallied for survivors’ three most important problems resulting from the bombing, and these were grouped in general themes of physical injuries and health problems; specific PTSD symptoms; other emotional symptoms; work, academic, financial and housing problems; and problems with interpersonal, family and friends. Additionally, tallied were responses to open-ended questions on survivors’ three current most important needs. These responses were also grouped in similar general themes of physical and health care needs; emotional issues; work, academic, financial and housing problems; interpersonal, family and friends matters; religion; and being positive or helping others.

Qualitative findings of problems and needs were discussed in relation to quantitative findings.

## 3. Results

Participants surveyed in this study included 138 OKC bombing survivors who were directly exposed to the event, of whom 45 (32.6%) were in the Federal Building during the blast and 93 (67.4%) were in nearby buildings just outside the building or in their cars. Medical treatments had been received by 112 (81.2%). Community controls included 171 individuals not exposed directly to the blast and who did not experience the death of family or close friends from the bombing.

Survivors and controls were demographically matched and did not differ in age (mean = 58.7 and 59.2 years respectively), gender (females = 52.2% and 52.6% respectively) or ethnicity (whites = 86.2% and 82% respectively). The groups differed in highest education level achieved; the survivor group had a significantly higher proportion of college graduates (χ^2^ = 9.62; df = 1; *p* = 0.0019) and a lower proportion of high school graduates/GED (χ^2^ = 11.16; df = 1; *p* = 0.0008) compared to controls. The survivor group also had a significantly higher proportion of married/member of an unmarried couple (χ^2^ = 6.48; df = 1; *p* = 0.0109) and a significantly lower proportion of divorced/separated/widowed (χ^2^ = 9.12; df = 1; *p* = 0.0025) participants compared to controls.

HSCL-25 anxiety scores and HSCL-25 depression scores were significantly higher in survivors than controls (*p* = 0.0195 and *p* = 0.02 respectively). See Table 1. Among survivors, 23.2% reported 4 or more PTSD symptoms on Breslau’s PTSD screen, consistent with a DSM-IV diagnosis of PTSD. 

Furthermore, for seven of ten posttraumatic growth (PTG) items, more than 30% of survivors reported PTG to a “great” or “very great” degree. These items were: changing priorities about what is important in life, having a greater appreciation for the value of my life, having a better understanding of spiritual matters, knowing better how to handle difficulties, having stronger religious faith, being stronger than I thought and learning how wonderful people are (Figure 1).

The incidence of seven major medical problems in the previous 12 months did not differ between survivors and controls in questions from the Medical Status Questions (MSQ) used in the OSDH’s prior follow-up survey (Figure 1). In addition, survivors and controls did not differ in reported emergency room (ER) visits or hospital stays, doctor’s office visits or mental health care utilization with a psychiatrist, psychologist, psychiatric nurse or clinical social worker in the preceding 12 months (Table 2). However, survivors reported significantly more visits to a physical, speech, respiratory or occupational therapist, considered as ancillary health care workers in this study. (Table 2).

For open-ended questions, 114 survivors provided at least one response to the question about bombing-related problems and 79 provided at least one response to the question about needs. (Table 3) For the question about survivors’ most important problem resulting from the bombing, physical injuries and health problems were endorsed by a total of 80 survivors (especially hearing difficulties for 22). Specific PTSD symptoms were noted by a total of 75 survivors. Other emotional problems were reported by a total of 72 survivors. Work, academic, housing or financial problems were reported by a total of 29 survivors and interpersonal problems were noted by 24. For the question about their most important current need, physical and health care needs were reported by a total of 40 survivors (including hearing-related issues for 10). Diverse emotional needs were noted by a total of 56 survivors. Work, academic, housing or financial needs were discussed by a total of 22 survivors. Interpersonal matters (including closeness to others and family issues) were among current needs for a total of 33 survivors. Religion was a need for one, and a total of nine survivors reported a need to be positive, give back or volunteer. Responses for the first, second and third listed most important problems and needs are presented in Table 3.

## 4. Discussion

Our results demonstrated both some similar and some differing pictures of long-term effects of the OKC bombing when assessed through questionnaires and open-ended interviews. Findings demonstrate the diversity of lasting sequelae that remain to be addressed after intense exposure to terrorism in a group with a high rate of injury (approximately 80% of our sample).

Quantitative comparisons of questionnaire assessment of long-term symptoms of anxiety and depression yielded significantly higher rates of both symptoms in survivors compared to controls on HSCL-25. However, it is not known whether these reported symptoms that are statistically different between the groups are clinically significant, that is, whether survivors suffered from higher rates of diagnoses of depressive or anxiety disorders. These reported anxiety and depression symptoms in direct survivors of terrorism on HSCL-25 are consistent with unprompted [qualitative] descriptions of emotional symptoms reported in open-ended questions 72 times for bombing-related problems and 56 times for current needs.

Of note, approximately 23% of survivors endorsed symptoms consistent with probable PTSD on the Breslau screen. This long-term PTSD was consistent with PTSD or PTSD symptoms reported spontaneously 75 times in open-ended questions and of bombing-related problems.

Other bombing-related problems and needs emerged from open-ended questions that were not encompassed by structured questionnaires. Thus, we learned from individuals of their work, academic, financial and housing problems for 29 in bombing-related problems and for 33 among their current needs. Additionally, problems with interpersonal matters were reported by 24 and similar needs by 33.

What the formal assessments of psychiatric problems also missed were the poignant expressions of individuals’ verbalized reports of bombing-related problems: “Psychophysiological reaction from a new section chief making a statement that ‘Hell, everybody knows the FBI blew up the Murrah building so they could get their budget from Congress.’” PTSD symptoms were intense: “Every time I hear a loud noise I jump.” “I think probably the one that I can’t control is the response to loud, sudden noises I am not expecting or someone coming into the room and turning a light on or the power off when not expecting it. I am more alert to the people around me…” “Just a general feeling whether you’re walking down the street, driving in your car, or sitting in your house that something could happen that instantly changes your life. Things can happen to you and really bad things…” The theme of loss after so many years emerged: “Just dealing emotionally with the loss of so many friends.” Among continued emotional needs were: “Feeling safe, safety with everything around me, especially going into restaurants; I’m very fearful about restaurants, a lot of people. I don’t go shopping anymore, I tend to stay away from crowds because I have to turn my head to look at everyone.” “I need my family with me, I need to feel secure, I need to feel safe and secure.”

Questionnaire assessment of seven major health conditions and general health care utilization in terms of ER visits, hospital stays and doctor visits in the previous 12 months did not differ between survivors and controls. It is not known if there are differences in severity of health problems reported by both groups, or whether their health problems and health care utilization are related to age, given mean ages of 58.7 years in survivors and 59.2 years in controls. However, these similar rates of reported health issues do not reveal specific long-term health problems and health care needs attributed to the bombing as discussed 80 times and 40 times respectively. These verbalized health care problems included hearing problems, musculoskeletal problems, pain, head injuries and eye or visual loss from the bombing. “I had a head injury so without a botox shot to my throat I can’t talk and until a month ago I had not been able to talk for over two years and it’s been like this for 15 years.” “I lost my right eye because of a migraine and I just developed migraines as a result of the bombing. I lost my right eye because of that and I stopped working. That’s the biggest change.” “Weakened lung and a breathing condition, chronic bronchitis, scarring in my lung, pneumonia this last winter.” “I had a piece of glass go through my elbow, so the lower part of my left hand has no feeling in it. So that is a physical limitation as far as holding stuff. My left hand is impaired somewhat.” Continued health care needs of various types were also reported by 40 survivors. “If I could have anything I wanted, it would be good health and not to be in pain.” “I would like to see a specialist or something for my lower back or for a brace for something for me to put on.” “Fix hearing; this is a major problem and continues to get worse—ringing in the ears is horrible.” Additionally, the need for emotional closeness emerged: “To feel connected with other people.”

Survivors’ greater use of ancillary health care (physical, speech, respiratory and occupational therapy) compared to controls may reflect lingering problems from bombing-related injuries that were reported spontaneously and in some detail on open-ended questions.

Posttraumatic growth in terms of using positive coping skills was reported by many survivors on PTGI-S, consistent with a few spontaneously reporting the need to become positive or help others. As they stated: “I need to work with the memorial to try and teach people that it is a horrific thing to happen, but that you can turn things into a positive.” “To try to have a positive impact on the lives of others, because the things you do for yourself has a pretty short shelf-life, and doing things for others is more impactful.” “I have the need to give back.” This attempt to cope by being positive and helping others after difficult life experiences and in the face of emotional distress has been noted in other terrorism survivors. A study of coping strategies used by Israeli mothers having prolonged exposure to missile attacks found that the PTSD symptoms were associated with greater use of problem-focused coping and greater PTG [19]. Similarly, a study of employees present at work during the 2011 Oslo bombing attack found that PTG was associated with greater posttraumatic stress [20]. This seemingly counter-intuitive association reflects individuals’ efforts to adjust by finding meaning and positive coping after adversity.

Limitations of this study include the fact that only 138 (33.9%) of 407 available survivors from previous research were surveyed, affecting generalizability. We do not know how many participants in the original group may have died since the original survey or have been unable to respond. Additionally, survivors and controls differed in educational level and marital/relationship status, possibly affecting group comparisons. We did not measure the severity or cause of medical problems reported in the MSQ in this relatively brief telephone survey, which may be relevant to the failure to find differences between survivors and controls in reported medical problems and general health care utilization. However, as stated, survivors’ increased use of ancillary health care services may have reflected continuing bombing-related problems and unmet health needs noted in open-ended questions. Interviewers were not blinded to respondents’ bombing exposures, which may have affected how questions were asked and documented. Moreover, in asking open-ended questions it is possible that these questions or the way they were posed may have implied to survivors that such problems were expected, leading to bias. In our brief survey, controls were not asked about their most important problems or needs in order to determine whether survivors’ responses attributed to the bombing were not unique to their terrorism exposure. In addition, grouping of responses to open-ended questions may miss overlapping themes. Finally, not all survivors chose to respond to open-ended questions, for unknown reasons, so that some problems and needs may have been missed.

## 5. Conclusions

Despite the above limitations, our study suggests that direct survivors of terrorism experience a diversity of long-term health and mental health problems and needs which may differ in intensity and subjective experience according to the type of assessment (questionnaire versus open-ended interview) used. Reliance on only questionnaires may miss many issues that survivors may voice when invited to discuss their personal stories. Results suggest that extended recovery services are needed long after terrorism exposure and that open-ended interviews provide valuable information to supplement questionnaire assessment in identifying those in need of help.

## Figures and Tables

**Figure 1 behavsci-11-00019-f001:**
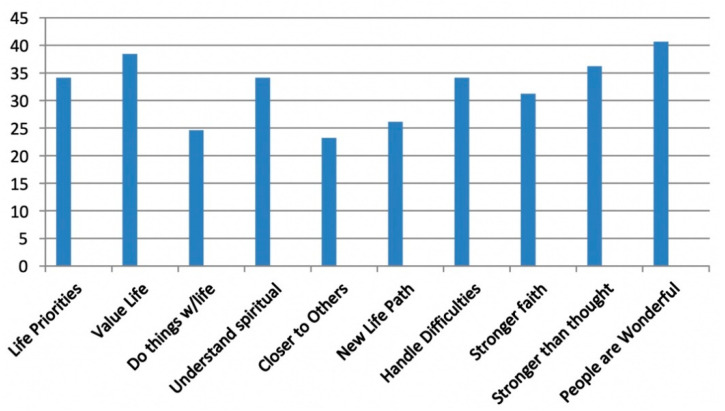
Posttraumatic Growth Inventory: % of Survivors Reporting Change to a Great/Very Great Degree after Bombing.

**Table 1 behavsci-11-00019-t001:** Anxiety and Depression Symptoms in Survivors and Controls.

Assessment	Group	Mean	SD	Mean Diff.	t	df	*p*
Hopkins Symptom Checklist (HSCL)-Anxiety	Bombing Survivors	15.7	5.3	1.4 (0.2; 0.6)	2.35	294	0.0195
	Controls	14.3	4.9				
HSCL-Depression	Bombing Survivors	24.3	9.4	2.4 (0.4; 4.4)	2.34	252.9	0.0202
	Controls	21.9	7.7				

**Table 2 behavsci-11-00019-t002:** Bivariate and multivariable analysis: Medical Diagnosis and Health Care Received.

Diagnosis	Survivors Total = 138 (%)	Controls Total = 171 (%)	Survivors Versus Controls Crude Odds Ratio (95% CI) Chi-Square/Fisher’s Exact Test	Adjusted Odds Ratio (95%CI) ^a^
Ever told by a doctor that you had a stroke	3/137 (2.2)	5/163 (3.1)	OR = 0.71 (0.17; 3.02) Fisher’s exact test: *p* = 0.7312	OR = 1.05 (0.21; 5.15); *p* = 0.9512
Coronary heart disease during past 12 months	9/137 (6.6)	10/162 (6.2)	OR = 1.07 (0.42; 2.71) χ^2^ = 0.02; df = 1; *p* = 0.8886	OR = 1.17 (0.42; 3.24); *p* = 0.7598
Had hypertension in the past 12 months	62/138 (44.9)	59/162 (36.4)	OR = 1.42 (0.90; 2.26) χ^2^ = 2.24; df = 1; *p* = 0.1344	OR = 1.56 (0.93; 2.61); *p* = 0.0901
COPD during the past 12 months	28/138 (20.3)	24/163 (14.7)	OR = 1.47 (0.81; 2.69) χ^2^ = 1.62; df = 1; *p* = 0.2031	OR = 1.53 (0.79; 2.96); *p* = 0.2071
Ever told by a doctor that you had diabetes	25/138 (18.1)	27/163 (16.7)	OR = 1.11 (0.61; 2.03) χ^2^ = 0.13; df = 1; *p* = 0.7227	OR = 1.42 (0.74; 2.74); *p* = 0.2966
Ever told by a doctor that you had cancer or a malignancy of any kind	20/138 (14.5)	28/162 (17.3)	OR = 0.81 (0.43; 1.52) χ^2^ = 0.43; df = 1; *p* = 0.5110	OR = 0.80 (0.41; 1.56); *p* = 0.5072
Ever told by a doctor that some form of joint problems, arthritis, rheumatoid arthritis, gout, lupus, or fibromyalgia	74/138 (53.6)	73/163 (44.8)	OR = 1.43 (0.90; 2.25) χ^2^ = 2.34; df = 1; *p* = 0.1264	OR = 1.61 (0.97; 2.68); *p* = 0.0683
Seen or talked to a physical therapist, speech therapist, respiratory therapist, or occupational therapist	33/138 (23.9)	24/163 (14.7)	OR = 1.82 (1.02; 3.26) χ^2^ = 4.11; df = 1; *p* = 0.0426 ^a^	OR = 2.03 (1.08; 3.82); *p* = 0.0285 ^a^
Seen or talked to a mental health professional such as a psychiatrist, psychologist, psychiatric nurse, or clinical social worker	23/138 (16.7)	19/164 (11.6)	OR = 1.53 (0.79; 2.94) χ^2^ = 1.62; df = 1; *p* = 0.2036	OR = 1.64 (0.78; 3.42) *p* = 0.1910

^a^ Adjusted for gender, ethnicity (Caucasian versus Other), age, marital status, and education level.

**Table 3 behavsci-11-00019-t003:** Most Important Bombing-Related Problems and Current Needs. 114 survivors had at least 1 response to questions. Responses are grouped according to general themes.

Bombing-Related Problems	#1 (*n* = 114)	#2 (*n* = 94)	#3 (*n* = 72)	Current Needs	#1 (*n* = 79)	#2 (*n* = 50)	#3 (*n* = 32)
Physical Injuries and Health Problems	37 Hearing = 11	33 Hearing = 10	10 Hearing = 1	Physical and Health Care	26 Hearing = 9	12 Hearing = 1	2
PTSD or PTSD Symptoms	29	25	21	Emotional	21	25	10
Other Emotional Symptoms	27	21	24	Work, Academic, Financial and Housing	12	6	4
Work, Academic, Financial and Housing	12	8	9	Interpersonal, Family, Friends	18	4	11
Interpersonal, Family, Friends	9	7	8	Religion	0	1	0
				Positive, helping others	2	2	5

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
