# Peer review of "Problems and Needs Persist for Oklahoma City Bombing Survivors Many Years Later"

_behavsci, 2021, doi:10.3390/bs11020019_

Round 1
Reviewer 1 Report
This is an interesting telephone assisted survey of reported symptoms and problems in OKC bombing survivors compared with matched controls.
The following issues need to be addressed:
1) in the Introduction some basic info about the OKC terrorist attack need to be given
2) ethical approval is not mentioned in the Methods section, authors refer to ref 9 for further info on methods but I think that ref 9 is misreported (I think first author is missing)
3) Authors give importance in the discussion to the differences found between the two groups in the level of symptoms reported in rating scales but they could as well give importance to the many not statistically significant differences found. Also, we do not know if the differences found are clinically significant, given the methods used
4) in the limitations Authors fail to mention the possible effect of the suggestive nature of the open questions. Asking to mention "the three most important problems experienced as a result of bombing" appeared inherently biased implying that such problems exists (or at least it suggests that they may)
5) also in the limitation section it should be noted that these open ended questions were not apparently asked to controls. Why? True, Authors could not ask to mention the three most important problems experienced as a result of bombing to controls but could have asked the three most important problems currently faced. Certainly they could have asked controls their most important needs. This would allow to have more strenght in concluding that the answers of the open questions capture something specific or unique to the OKC bombing survisors. It may be simply that asking open ended questions understandably allows the possibility to mention somethig that rating scales do not allow, this for ANY population. This does not mean necessarily that it is useful to ask those questions, that it is feasible, cost effective, that it is appropriate (in terms of risk or re-traumatisation), to ask those questions and conclude needs for intervention from the answers, whose clinical significance is unknown
Author Response
The following issues need to be addressed:
- in the Introduction some basic info about the OKC terrorist attack need to be given.
From authors: We have added basic info on the OKC bombing to the introduction, starting in line 58.
- ethical approval is not mentioned in the Methods section, authors refer to ref 9 for further info on methods but I think that ref 9 is misreported (I think first author is missing)
From authors: We added more on IRB approval to lines 100-101. Reference 9 (now reference 12 since 3 references were added) has been corrected. Thank you for noticing that.
- Authors give importance in the discussion to the differences found between the two groups in the level of symptoms reported in rating scales but they could as well give importance to the many not statistically significant differences found. Also, we do not know if the differences found are clinically significant, given the methods used.
From authors: We have added more discussion on the importance of the non-statistically significant differences found—specifically in the areas of major medical problems and general health care utilization, for which open-ended questions revealed major bombing-related long-term medical problems and needs and more ancillary health care utilization that might result from bombing injuries. We also discuss work, academic, financial and housing problems and interpersonal matters that emerged in open-ended questions that were not covered by formal surveys.
- in the limitations Authors fail to mention the possible effect of the suggestive nature of the open questions. Asking to mention "the three most important problems experienced as a result of bombing" appeared inherently biased implying that such problems exists (or at least it suggests that they may)
From authors: We have added this to the limitations section.
- also in the limitation section it should be noted that these open ended questions were not apparently asked to controls. Why? True, Authors could not ask to mention the three most important problems experienced as a result of bombing to controls but could have asked the three most important problems currently faced. Certainly they could have asked controls their most important needs. This would allow to have more strength in concluding that the answers of the open questions capture something specific or unique to the OKC bombing survivors. It may be simply that asking open ended questions understandably allows the possibility to mention something that rating scales do not allow, this for ANY population. This does not mean necessarily that it is useful to ask those questions, that it is feasible, cost effective, that it is appropriate (in terms of risk or re-traumatization), to ask those questions and conclude needs for intervention from the answers, whose clinical significance is unknown.
From authors: Good points. We now discuss in limitations that in our brief survey we did not ask controls their important problems and needs to determine if controls’ problems and needs were unique.
We have also rewritten several sections of our manuscript for more clarity.
Reviewer 2 Report
I believe that if rewritten this paper could add valuable information to the literature regarding posttraumatic growth.
If I understand correctly this is information that was gathered some seven years previously and you are now wishing to publish with a different slant. This would be possible if you bring the focus onto posttraumatic growth, but in doing this you will need first to define this appropriately.
Since the information you have gathered is qualitative then obviously your paper needs to be restructured as such.
Some specific points include:
Problems and Needs Persist for Oklahoma City Bombing 2 Survivors Many Years Later
Bulky – needs rewording
ABSTRACT:
Overall, I think you need to rewrite this abstract as it comes over as bulky and rather unclear.
Lines 9-10 The focus of this study is on the possible lasting physical and emotional unmet needs of survivors of terrorism.
Line 10 Telephone questionnaire
Line 14 Is this 80% of the 138 persons.
It is very unclear - of the 138 persons directly exposed - were only 80% actually injured? When you say injured, do you imply physically? I would suggest that all 138 who were directly exposed would have some sort of injury - either physical or psychological.
KEYWORDS:
Posttraumatic growth: Please clarify what you mean by this term
INTRODUCTION:
Lines 29-31 References required for this statement.
Paragraph 2 Please rewrite in clearer terms.
Lines 43-48 Needs rewriting – many grammatical errors and does not read well
Line 56 Are you meaning one-half of the one third of survivors with PTSD? This is not clear.
Lines 55-59 Lots of numbers here and it is not clear what they are referring to.
Line 60 Does this relate to all survivors or only the ones with PTSD?
Lines 77-78 If this is the study you are now reporting on why have you used some of the results in your introduction? Also this study was effectively carried out in 2013 - why have you not published before 2021???
METHODS:
Statistical Analysis:
If this is a qualitative study using open ended interviews why are you not using qualitative analysis????
RESULTS:
I did not find the tables particularly helpful.
As I said earlier – this was a qualitative study and I would have preferred to see a qualitative type of analysis.
See paper for further comments.
DISCUSSION:
I do not think that the authors drew enough from the responses they received and I believe that this could have been in much greater depth.
Please see the attached paper for further comments

Author Response
ABSTRACT:
Overall, I think you need to rewrite this abstract as it comes over as bulky and rather unclear.
From Author: I have rewritten the abstract for clarity, and have given it formal structure.
Lines 9-10 The focus of this study is on the possible lasting physical and emotional unmet needs of survivors of terrorism.
From author: this is now succinctly stated in background of rewritten abstract.
Line 10 Telephone questionnaire--From author: done
Line 14 Is this 80% of the 138 persons.
It is very unclear - of the 138 persons directly exposed - were only 80% actually injured? When you say injured, do you imply physically? I would suggest that all 138 who were directly exposed would have some sort of injury - either physical or psychological.
From author: This is clarified--80% of 138 directly exposed survivors were physically injured.
KEYWORDS:
Posttraumatic growth: Please clarify what you mean by this term
From author: Posttraumatic growth is better defined in the methods section (PTGI-S scale), in the results section and in the discussion section (lines 168-173) and in the discussion section (lines 286-299).
INTRODUCTION:
Lines 29-31 References required for this statement.
From author: 3 new references are provided (new line 32)
Paragraph 2 Please rewrite in clearer terms.
From author: Paragraph 2 on international terrorism has been rewritten for clarity.
Lines 43-48 Needs rewriting – many grammatical errors and does not read well
From author: These lines and Paragraph 3 on domestic terrorism are rewritten.
Line 56 Are you meaning one-half of the one third of survivors with PTSD? This is not clear.
From Author: This is one half of the group of survivors (new line 62).
Lines 55-59 Lots of numbers here and it is not clear what they are referring to.
From Author: Hopefully this now shows in lines 61-61 that there was a slight decrease in rates of PTSD diagnoses over time in studies drawing from the same direct survivor registry.
Line 60 Does this relate to all survivors or only the ones with PTSD?
From Author: This does not relate to PTSD. See lines 67-68: "Also survivors’ anxiety and depression symptoms (but not PTSD symptoms) were associated with heavy drinking."
Lines 77-78 If this is the study you are now reporting on why have you used some of the results in your introduction? Also this study was effectively carried out in 2013 - why have you not published before 2021???
From author: Quite honestly, one of our authors (second author) believed more recently that this qualitative data should be reported. See new lines 87-88: We have added: “This study expands our understanding of long-term problems beyond our existing report of enduring problems noted above that used traditional rating scales 18 ½ years post-disaster [12].”
METHODS:
Statistical Analysis:
If this is a qualitative study using open ended interviews why are you not using qualitative analysis????
From author: See new lines 128-144. We have discussed in methods section that both quantitative and qualitative methods were used, and that "Qualitative findings of problems and needs were discussed in relation to quantitative findings." Our qualitative methods were consistent with
RESULTS:
I did not find the tables particularly helpful.
From author: Tables may be eliminated at the discretion of editors.
As I said earlier – this was a qualitative study and I would have preferred to see a qualitative type of analysis.
From author: see above
From author: The last paragraph of the results section (lines 178-188, now lines 200-215) has been rewritten to make data clearer.
DISCUSSION:
I do not think that the authors drew enough from the responses they received and I believe that this could have been in much greater depth.
From author: The discussion section has been provided with much more in-depth examination of both quantitative results on formal, structured assessments and qualitative results on open-ended questions—where they agree and where open-ended questions pick up on issues not emerging in formal survey items. Specifically we have added more discussion of work, academic, financial and housing problems and needs and interpersonal problems. We have also added more discussion of ancillary health care use findings and posttraumatic growth.
Also the limitations section has been expanded in several areas. We have also added to limitations that some aspects of open-ended questions may affect results (suggesting that problems and needs from the bombing do exist, inflating issues reported.)
Round 2
Reviewer 1 Report
Authors have satisfactorily addressed the issues raised providing a much improved manuscript.
Author Response
No suggested revisions were made. Thank you.
Reviewer 2 Report
Thank you for your changes this now reads much better. I have added my comments to the actual manuscript which I will attach.
One point I would like to stress - this additional material added to the introduction explains the reasons for this new publication.
However - it is still not clear if these open-ended questions were asked in the original telephone survey but the answers were never analyzed and published.
Please make this clear.
My other issue is that I prefer all numbers under 10 to be written in full and also all fractions to be written in full. So I have been possibly a little over-zealous in this!
However - greatly improved - thank you

Author Response
Thank you for your changes this now reads much better. I have added my comments to the actual manuscript which I will attach.
Authors: Thank you for your feedback. Hopefully we have understood and incorporated your suggested changes. Suggested changes to abstract were made, but word count is now 204, over the 200 word limit. Hopefully this is acceptable to editors.
One point I would like to stress - this additional material added to the introduction explains the reasons for this new publication.
However - it is still not clear if these open-ended questions were asked in the original telephone survey but the answers were never analyzed and published.
Please make this clear.
Authors: We have tried to clarify that the open-ended questions were asked in the original telephone survey. Hopefully this better clarifies it: “In this study we re-examine survivors’ responses to open-ended questions elicited at the same time that structured questions had been asked to compare qualitative with quantitative data. Thus, this study expands our understanding of long-term problems beyond our existing report of enduring problems noted above that used traditional rating scales 18 and a half years post-disaster [12].”
My other issue is that I prefer all numbers under 10 to be written in full and also all fractions to be written in full. So I have been possibly a little over-zealous in this!
Authors: We have replaced digits for numbers of ten or less and used terms such as “and a half” or “one third” instead of digits.
However - greatly improved - thank you
Authors: Thank you.